# Toward a Theory of the Underpinnings and Vulnerabilities of Structural Racism: Looking Upstream from Disease Inequities among People Who Use Drugs

**DOI:** 10.3390/ijerph19127453

**Published:** 2022-06-17

**Authors:** Samuel R. Friedman, Leslie D. Williams, Ashly E. Jordan, Suzan Walters, David C. Perlman, Pedro Mateu-Gelabert, Georgios K. Nikolopoulos, Maria R. Khan, Emmanuel Peprah, Jerel Ezell

**Affiliations:** 1Department of Population Health, New York University Grossman School of Medicine, New York, NY 10016, USA; maria.khan@nyulangone.org; 2Division of Community Health Sciences, University of Illinois at Chicago School of Public Health, Chicago, IL 60612, USA; lesliedw@uic.edu; 3Center for Drug Use and HIV/HCV Research, New York, NY 10003, USA; ejordanashly@gmail.com; 4Department of Epidemiology, New York University School of Global Public Health, New York, NY 10003, USA; suzanmwalters@gmail.com (S.W.); ep91@nyu.edu (E.P.); 5Icahn School of Medicine at Mount Sinai, New York, NY 10003, USA; david.perlman@mountsinai.org; 6Graduate School of Public Health and Health Policy, City University of New York, New York, NY 10027, USA; pedro.mateu-gelabert@sph.cuny.edu; 7Medical School, University of Cyprus, 2029 Aglantzia, Nicosia 1678, Cyprus; gknikolopoulos@gmail.com; 8Africana Studies and Research Center, Cornell University, Ithaca, NY 14850, USA; jme246@cornell.edu

**Keywords:** racism, people who use drugs, HIV, substance use, capitalism, scapegoat

## Abstract

Structural racism is increasingly recognized as a key driver of health inequities and other adverse outcomes. This paper focuses on structural racism as an “upstream” institutionalized process, how it creates health inequities and how structural racism persists in spite of generations of efforts to end it. So far, “downstream” efforts to reduce these health inequities have had little success in eliminating them. Here, we attempt to increase public health awareness of structural racism and its institutionalization and sociopolitical supports so that research and action can address them. This paper presents both a theoretic and an analytic approach to how structural racism contributes to disproportionate rates of HIV/AIDS and related diseases among oppressed populations. We first discuss differences in disease and health outcomes among people who use drugs (PWUD) and other groups at risk for HIV from different racial and ethnic populations. The paper then briefly analyzes the history of racism; how racial oppression, class, gender and other intersectional divisions interact to create health inequities; and how structural racism is institutionalized in ways that contribute to disease disparities among people who use drugs and other people. It examines the processes, institutions and other structures that reinforce structural racism, and how these, combined with processes that normalize racism, serve as barriers to efforts to counter and dismantle the structural racism that Black, indigenous and Latinx people have confronted for centuries. Finally, we discuss the implications of this analysis for public health research and action to undo racism and to enhance the health of populations who have suffered lifetimes of racial/ethnic oppression, with a focus on HIV/AIDS outcomes.

## 1. Introduction

HIV/AIDS and other diseases disproportionately harm racially and/or ethnically oppressed populations, including among populations of people who use drugs (PWUD) [1,2]. After discussing racial and ethnic differences in disease and health outcomes among PWUD and other groups at risk for HIV, the paper analyzes the structural racism that contributes to these disparities, including its history, its intersectional aspects, and how it takes institutional forms; some attention is given to international considerations, but the focus is on the United States, primarily for reasons of space. Structural racism also contributes to higher rates of many other diseases, morbidities, and death in non-drug-using oppressed populations. Thus, the focus of this paper is “upstream,” on structural racism and why it has not ended.

The paper then examines the processes and structures that have maintained racial, ethnic, and other oppressions over many centuries despite active efforts to eliminate such oppression [3,4]. After describing reasons why racism continues as an organized, multifaceted system, we conclude by discussing research and actions that might help eliminate racism as a system of unjust power and as oppression.

### 1.1. Description of Disparities

Structural racism affects HIV/AIDS in PWUD and other groups in many countries. In a meta-analysis in 2012 of data from 23 countries, Des Jarlais et al. found that people who injected drugs who were subjected to racial and/or ethnic oppression were more likely to become infected with HIV than were dominant, nonoppressed groups and that “among people who inject drugs, ethnic minorities are approximately twice as likely to be HIV seropositive than ethnic majorities” [5]. Moreover, Millett et al. compared prevalence rates among sexual minority men (SMM) in the US, United Kingdom, and Canada and found that Black SMM were more likely to be infected than non-Black SMM and that issues connected with structural racism were likely to explain the differences among SMM [6]. As far as we know, there have been no similar multination studies for HIV rates by race and ethnicity for female sex workers, heterosexuals, or other key affected populations. UNAIDS made an attempt to conduct such estimates in 2021 but concluded that doing so was beyond their available resources [7].

A great deal of information is available for the United States, so we will present this more specifically. We focus first on PWUD, often homing in on injection drug use, and then extend our presentation to the population as a whole. Higher rates of HIV, AIDS, and HIV-related mortality (e.g., Pneumocystis pneumonia) among Black and Latinx people who injected drugs (injection drug use was a prominent transmission route), as compared with rates among Whites, became visible in the United States (US) in the mid-1980s. In New York City (NYC), the initial epicenter of HIV among people who injected drugs in the US, these differences in HIV prevalence had emerged by the late 1970s and continue through the present for Black and Latinx people as well as other oppressed racial and ethnic minorities groups [8] (The data reported on in this paper were collected over a span of almost 50 years. During that time, the preferred nomenclature for different racial and ethnic groups has changed. In the early years, for example, the term “non-Hispanic” was not used in collecting data about Whites and Blacks. We have attempted to use terms that accurately reflect this. In addition, we use “Black” rather than any version of African and American because this is the term most often used when antiracist movements have been at their height, and “Latinx” rather than “Hispanic” in an attempt to be inclusive of those who occupy Hispanic and Latino/a/x identities or categories (see Vidal-Ortiz S and Martínez J [9]). However, we recognize that this term does not resonate with all Hispanic and Latino/a/x persons and that many do not use the term Latinx for themselves. Therefore, the nature of the term and usage may be problematic (see María Del Río-González [10])).

Structural racism is a system that oppresses non-White people such as Black people, Native Americans, and Latinx people as well as people who identify with multiple races and ethnicities [11]. This system has led directly and indirectly to these oppressed populations having much higher HIV rates than Whites (This is true for COVID-19 as well: Black, American Indian, and Hispanics/Latino/a/x people are more likely to get sick and die in the US. Their relative rates of hospitalization and of death (as compared with non-Hispanic Whites) as of 21 November 2021 were, for Blacks, 2.6 and 1.9; for Native Americans, 3.3 and 2.2; and for Hispanics/Latinos, 2.5 and 2.1, all respectively (Source: https://www.cdc.gov/coronavirus/2019-ncov/covid-data/investigations-discovery/hospitalization-death-by-race-ethnicity.html; accessed on 28 November 2021)). In 2019 among the general US population, HIV diagnoses among adults and adolescents per 100,000 persons were 45.4 for Blacks/African Americans, 20.4 for Latinx, 10.5 for American Indian/Alaskan Natives, 4.5 for Asian Americans, 18.2 for those identified as being of multiple races, and 5.3 for Whites [12]. These disparities are similar for HIV incidence and for AIDS mortality [13].

Racial and ethnic disparities in health and disease among the population as a whole long predated HIV and include differential rates of tuberculosis [14], syphilis and other sexually transmitted infections [15,16], cancer [17], diabetes [18], cardiovascular and respiratory diseases [19], obesity [18], fatal and nonfatal drug overdose [20], deaths and injury due to violence [21], and the ongoing COVID-19 pandemic [22,23,24]. This has been recognized as resulting from what has been called medical apartheid [25].

Given the breadth of racial and ethnic health disparities, their long-term duration, and the fact that these disparities have continued despite many efforts to reduce them, much of the rest of this paper focuses upstream on the history of racism and the forces that maintain structural racism. In this paper, we discuss (primarily) US-based racism against racially and ethnically oppressed PWUD, using data on Black and Latinx people generally as a case study of structural racism because these groups constitute >60% of HIV diagnoses in 2019 [26]. Moreover, we further explore how racism adversely impacts HIV and drug use epidemics in the US and point to implications about how this racism and its intersections with gender and class oppression contribute both historically and at present to ongoing global racial and ethnic oppression as a result of the political, military, cultural, and economic power of the US.

### 1.2. History

What are the social processes that created “race” and racism? And how did these get embodied into human beings in ways that led to these huge disparities in HIV/AIDS infection, sickness, and death? (We are using the concept of embodiment as developed by social epidemiologist Nancy Krieger (see [27,28,29]). It is a multidimensional concept that focuses on how life experiences, and the macro- and microsocial forces that shape experience, affect the individual- and population-level health of people. Thus, embodiment is the process by which social processes, social structures, and biography interact with bodies over time, positively or adversely impacting health in various ways).

Racism, the concept of race, and the embedding of this concept into social and cultural institutions are all historical constructs [30,31]. Whatever ideas about human inequality existed prior to capitalism, these ideas, and the social relationships that helped them seem believable, were substantially remade in the course of the European (and later US) invasion, conquests, and colonial control of much of the Americas, Africa, and parts of Asia [32,33]. White slave owners, slave traders, and European investors in the slavery systems, and others, developed propaganda, books, and pamphlets to disseminate papally-sanctioned racial ideas about “Africans as irredeemably inferior and perhaps not even human” that propped up the human slave system and the profit-oriented economy that grew up with it [34,35]. Although this section of the paper focuses mainly on history, and on the oppression of Black people in particular, it is important to realize that modes of racialized exploitation and violence continue to be re-produced through seemingly new systems of structural racist violence. One of these systems is foreign “aid”, such as World Bank loans with structural adjustment programs that require that nations assume debt to foreign lenders and ensure that funds are made available to repay these lenders by cutting funding for public services [36]. These modes are new expressions of structural racist legacies embedded in colonialism, the global human trafficking/slave trade and slave system, and the capitalist/imperialist expansion that has been producing and re-producing race and ethnicity-based oppression and exploitation for centuries [37,38,39]. Here, one manifestation of this history of structural racism in the history of the AIDS epidemic was the early appearance of HIV in Haiti in the late 20th century; Haiti had been underdeveloped by the slave system and subsequently by forced indebtedness, and when the concept of HIV “risk groups’ was being developed, Haitian identity was initially labeled a risk [40].

The colonial pattern of having White rulers and having Black and/or other oppressed people doing manual and other work was consistent across multiple geographic locations and tended to strengthen and support racist ideas in all cases. Examples include Spanish elites forcing indigenous and Black peoples to work in silver mines and fields in South America, Dutch elites dominating peasants and others in what eventually became Indonesia, and the slave trade and plantation slavery that underlay the sugar manufacturing and trade in the West Indies that was a major part of capitalist development for many decades, and the slave-based cotton economy that later propelled US economic development [41,42,43].

During this period, some Europeans also worked in the fields or in other lower-level positions in some locations, in many cases as prisoners or under contracts of indenture that bound them to work as repayment for their passage overseas. This pattern posed the possibility that multiracial rebellions could develop and perhaps unseat the power and profits of these elites. Theodore Allen has analyzed one important case of such a revolt, Bacon’s Rebellion in Virginia in 1676–1677, and showed how rulers’ reactions to this struggle led the elite to create privileges for White workers and to stigmatize Black people as part of a conscious and deliberate divide-and-rule strategy. Importantly, some groups who had been oppressed in other settings by divide-and-rule strategies were able to become incorporated into the US racial system as “White” with moderate levels of struggle following migration to the US [44,45]. For example, Irish people, who had been colonized and racially oppressed by Britain, were able to overcome initial ethnic oppression in the US by adopting anti-Black racist attitudes and behaviors and “becoming White” [46].

Taken together, racial categorization and division to weaken potential solidarity among the laboring classes, racial subordination, racial stigmatization, mass incarceration of oppressed groups, and a crystallizing ideology of White supremacy help constitute the individual, social, and institutional components of racism in the current era. They tend to some degree to be self-perpetuating, as people of various classes, genders and races and ethnicities come to accept them as part of the “natural order”. On the other hand, they are also strengthened by deliberate actions on the part of the rich and powerful as part of their efforts to maintain their wealth and power, in corporate and asset ownership and control [47], in health care systems [48], in media systems [49,50], in local and state, and national governments, and in the world as a whole [51,52,53,54,55,56].

One example can usefully illustrate this, as well as the way in which the racist divide-and-rule strategy of the ruling classes came to shape the ideologies of these same classes and then to spread these ideologies and related organizational techniques to other classes in the US and internationally. Esch and Roediger [57] describe how managerial practices in the US created employment and workplace realities in the 1800s and early 1900s that helped maintain a racialized culture and intersectional divisions in the working class. Employers and their managers set up highly racialized job classification systems that assigned people to jobs according to racial, ethnic, and gender categories. They intended this to prevent workers from uniting by giving them different jobs with different work patterns, prestige, and pay levels (and hence to some degree different real and apparent interests), and having supervisors treat them differently by race, ethnicity, and gender in conformity with employers’ and managers’ racist and sexist beliefs.

The experience of living and working in unequal conditions itself came to seem natural (as mentioned above), and this still perpetuates and reinforces racist beliefs and cultures among many workers, both White workers who adopted or developed varying degrees of racist beliefs and workers of color who may have developed varying degrees of internalized racism. Further, newspapers often published managers’ ideas on which groups lacked or which groups had intelligence, dexterity, and other abilities.

These approaches spread internationally, as also discussed by Roediger and Esch [55]. For example, as the US grew in power, many US mining companies replaced White European managers with White American managers in mines in Asian, Mexican, South American, Australian, and Africa. This was done due to the Americans’ “expertise” in “race management”. Herbert Hoover (later President of the US) became wealthy and famous as an international consultant who could increase mining profitability by using these racist methods. Henry Ford, a noted anti-Semite and supporter of fascism, applied these techniques to his company’s foreign operations [58]. On an ideological level, Frederick Taylor’s “scientific management” ideas, which were extremely influential in elite efforts to control labor (what is euphemistically referred to as “labor relations”) internationally, incorporated detailed specifications of the putative “talent” or “lack of talent” of different racial and ethnic groups for different tasks [59,60].

These racialized and racist systems and process have contributed to vast racial/ethnic disparities in HIV disease distributions among people who inject drugs, sexual minority men, heterosexual women, and others, with HIV infection concentrated in urban, racially segregated “hot spots” [61].

## 2. Intersectionality Developed as a Parallel and Reinforcing Historical Process

Although we do not have space in this article to do the subject justice, at the same time that colonial conquest and capitalism fostered the development of a new system of racial and ethnic domination and its related ideologies, the colonial conquests and the development of capitalism also reshaped family structures, gendered work patterns, and gender power relations and ideologies [62,63,64]. This led to complicated patterns of diverse experiences and interests that provide fertile soil for elite divide-and-rule strategies and, indeed, for cross-category disagreements and hostilities without active elite intervention [65].

The intersection of class, race and gender, and its usefulness for employers and government elites’ efforts to maintain their wealth and power, further specifies the discussion in the history section on race/ethnicity and its uses for the powerful. This can be illustrated through some events in Detroit, Michigan, for many decades the center of the US auto industry. From the outset, the auto companies hired White males almost exclusively as their workers. After the packinghouse and steel strikes of 1919 revealed additional ways racial divides could be activated as an anti-union strategy, and the revolutions in Russia, Germany, and other European countries underlined the reality of the threat to capitalist power, auto employers began to hire Black men as a hedge against labor unrest [66]. This worked for years, but interracial solidarity among the male workers grew to be sufficient to overcome this during the labor upsurge of 1936–1941 [66]. During World War II, many of the workers who won these victories were in the military, so the auto companies hired workers with no union experience (many of them Whites from particularly racist parts of the US South), many women, and considerable numbers of Black men and women. This created the basis for White hate strikes against Black peoples, even though it also provided the basis for massive wildcat strikes against company rules, working conditions, and other issues [67]. The experiences of Black women auto workers at this time are described by Charles Denby in his autobiography, in which he describes their dual oppression and exploitation: “all the White women sat with the White men. They were always mixed with the men and didn’t eat with us”; regarding actual labor roles: “the White women had the light work”; and regarding opportunities indicating that White women workers were prioritized over Black women workers even if Black women workers were more “experienced”. The multiple and interacting oppressions of being both Black and woman were reinforced and perpetuated by employers and managers who systematically relegated Black woman workers to the lowest-paying and most labor-intensive and dangerous roles compared with their White woman counterparts [68,69].

These intersectional racial, ethnic, and gender-based oppressions among laborers mirror the processes that contribute to oppressions among HIV-affected populations. An intersectional framework also helps us understand the distributions of health risk and morbidity/mortality at both the individual and population levels. Importantly, within racial, class, gender, and sexual minority categories, populations experience different risks of acquiring HIV and, if infected, of obtaining quality, effective care (Intersectionality can be described an analytical framework that challenges the notion that multiple social stratifications of power exist (or can be studied) as independent entities. It is based on a belief that socially and economically constructed categories have an element of interdependence and inter “activity” in peoples’ lives that can potentially both cause and increase marginalization in a given society. Using intersectional approaches, we can explore how the different aspects of people’s lives (e.g., social, economic, educational, cultural) relate to each other and how they are positioned in the hierarchies of power in a given society. This enables us to critically assess the resultant impact on people’s access to resources and services within their communities and in turn, the impacts of these “intersectional” aspects of living in society on the life chances of those marginalized in society (see [70])). These conditions include experiencing multiple stigmas and vulnerabilities due to identifying oneself as a PWUD or being identified by others as a PWUD interacting with the experiences of stigma and other oppressions due to sexual minority status. These oppressions affect the prevalence of poverty, unemployment, police surveillance, incarceration, educational opportunities, sexual violence, and child abuse [71,72]. Further, PWUD with HIV, sexual minority PWUD (women who have sex with women and SMM, in varying ways), Black PWUD, and Black women PWUD experience overlapping stigmas and oppressions.

These intersectional differences affect HIV prevalence among PWUD, in part by influencing who has sex with and who injects with whom—that is, by shaping risk networks intersectionally. Kottiri et al. showed that in NYC during the early 1990s, the tendency for black injectors’ networks to include more black injectors (together with black PWUDs’ higher HIV prevalence) helped explain their higher probability of being HIV-infected [73]. In addition, the intersectional socially and economically produced conditions adversely impacting PWUD of different genders and sexual orientations, together with network characteristics, help explain why women who inject drugs and have sex with women are more likely to become infected with HIV than heterosexual women, and why they also tend to become infected within fewer years after injection initiation than do men [74,75].

## 3. Institutionalization

As discussed above, one way to analyze racism is as a system that developed historically in a process driven by strategies of the rich and powerful to maintain labor and social relationships established during the period of conquests, slavery, and the White expropriation and occupation of indigenous lands. This racist system helped them maintain their power and profits, a process assisted by the fact that patterns of domination and stratification seem “natural” to those who benefit from them and live within them as well as to some who are oppressed.

Another, complementary way to analyze racism is to look at how Black (and other) racial subordination is both institutionalized and a major part of many cultures and ideologies. In this section of the paper, we will discuss the structure of such institutionalization. The nature of such structured and organized racial subordination is shaped by the fact that both the racially oppressed and other people often resist it. Thus, maintaining and enforcing a system of racial subordination requires social roles in which the rules lead officials or other role-holders (who may or may not themselves knowingly hold racist ideas) to act in ways that disadvantage and disempower members of oppressed “races.” In practice, although many people become social workers or police officers while holding anti-racist ideas (particularly if they are members of oppressed groups), the daily experience of enforcing rules that are fundamentally based on racist assumptions, which are implemented using racist protocols or procedures, or which have racist effects, leads such people either to abandon these roles or to begin to adopt elements of racist perspectives themselves [76].

Maintaining and enforcing a system of racial subordination also requires systems of education and propaganda that promulgate and reinforce the ideas that there are races and that the oppressed racial groups are not competent or motivated and further, are not “moral” or even human and are therefore more likely to do evil things. Examples include sitcoms that feature police arresting or shooting Black “perps,” and mainstream news coverage of a “solemn field of blue” paying a respectful and “tearful farewell” to a police officer, even a Latinx one, killed in Harlem [77]. Education systems in the US present the history of US massacre, conquest, and expropriation of indigenous tribes and of Mexican territories in a positive light, and they almost never discuss their past and ongoing oppression [78,79,80]. Similarly, schools’ treatment of Black struggles and oppression is scant and misleading, and efforts to expand it often face fierce resistance [81]. All of this strengthens and inculcates racist ideas that, when held by millions of people, are themselves a powerful support of the racists system and thus are a barrier to popular resistance to racism or to the rule of the wealthy and powerful.

Such reinforcement of racist beliefs has been strengthened by the ongoing monopolization of mass media in formats such as television, the internet, social media and other “news” platforms. Since many media companies are owned by the same owners, or others who share similar ideas, messages easily get spread repeatedly to other stations or platforms seemingly independent from each other [82,83]. Artificial intelligence (AI, e.g., “Twitter bots”) further contributes to the ease of which racists messages can be propagated on various platforms [84], leading to the concept of “platformed racism” [85]. Misinformation on online platforms has emerged during times of crisis or elections for example after the Sandy Hook school shooting, the 2016 and 2020 US Presidential elections, and the COVID-19 pandemic [86,87,88].

Racism itself is institutionalized in many ways as an organized system of power. Critical race theorists and others have emphasized the extent to which racism has been codified in the US Constitution and in federal and state laws and administrative regulations such as those that encourage redlining and other forms of housing segregation, education financing systems that are localized and that thus underfund schools attended by Black, Latinx and other oppressed groups, and the destruction of the neighborhoods in which the oppressed live [56,89]. Racism is embedded in the very basic level of the structure of the US as a federal system, including the requirement that legislature be approved by both a more popularly elected House and a more elite Senate. These structures were explicitly intended (as stated by Founding Fathers Hamilton, Madison, and others) to weaken social movements’ (“factions”) ability to weaken the control of the rich and powerful White plantation owners, merchants, and owners of capital who have led the US government since its inception. Originally, the racism was overt, as in clauses that supported the slaveholders, and in a Senate appointed by state legislatures. The Senate, even after becoming an elected body, has consisted overwhelmingly of rich, White men and has functioned potently to support chattel slavery, Jim Crow segregation, and the weakening of unions and organizing efforts [90].

However, the total system of racism involves not only laws but also formal organizational hierarchies that enact it at national, state, and local levels that devise and promulgate scapegoating and other ideological tropes that justify, motivate, and normalize the actions of racist institutions and organizations and that set up rules and regulations that lead workers in a vast number of private and public organizations to treat White people differently from Black people and other oppressed groups. Racism is also continually reinforced by both less official formal organizations and less formal organizations, as well as a degree of popular support among (primarily) White populations and to a lesser degree among some of the oppressed [78,91,92]. Less official (but still formal) organizations, often organized nationally and hierarchically, have included the Ku Klux Klan, the White Citizens Councils, the Texas Rangers (which violently oppressed indigenous peoples and aggressively sought to recapture Black people who had escaped from slavery, and currently include White neighborhood watch groups and White anti-Latinx vigilante groups at the U.S.-Mexico border with implicit or explicit support by White local, state and federal governments), various current armed vigilante groups and militias, and informal street gangs that violently try to keep oppressed groups out of their neighborhoods [78,93,94]. They also have included organizations like the Partnership for a Drug-Free America that have helped implement and support racist policies that appear on their face to be color blind and indeed often garner support from among more established and wealthier sections of oppressed racial and ethnic groups. Organizations of these kinds have historically included racist vigilante groups, which have in turn emboldened and facilitated the growth of other racist vigilante groups in the US [95]. Some online White supremacist organizations act as hacker collectives to harass and stifle online communication among racial and ethnic minority communities [96,97,98].

Although specific patterns vary internationally and historically over time, capital and governments have been closely intertwined within almost every country for many generations. There is a vast literature on the extent to which this is true and the exact degrees and procedures through which it has been organized and through which the very real disagreements that take place among political and economic elites are resolved [99,100,101,102,103]. Since our primary focus in this paper is the US, most of our citations above deal with that country. The extent to which other countries’ policies support racism, and the kinds of racism they support, also vary over time. The continued economic and political oppression of most African, Latin American and Caribbean, and Asian countries, and of peoples in these countries, by the White powers (and by other racist state powers such as China and Japan) shows the extent to which racism takes international forms [104,105,106,107,108]. Within many of these countries, it should be added, various forms of racial and ethnic oppression also occur, including various forms of racism based on shades of perceived skin color (often called “colorism”), rather than pure Black/White dichotomies [109].

This institutionalized racism affects HIV and other diseases. One clear example is the War on Drugs, which was announced and enforced as a reaction to the social movements challenging racism in the 1960s, and the way media and some churches and politicians reacted to HIV/AIDS among PWUD. This reaction produced strong negative reactions to HIV prevention programs such as syringe exchanges in the late 1980s and early 1990s. These attacks delayed and limited the establishment of these programs and thereby contributed to a great many people needlessly becoming infected with HIV [110,111].

## 4. Embodiment

The result of these racist structures and ideas for the HIV/AIDS epidemic and COVID-19 pandemic have been severe, as shown by the statistics given above. The ways in which racism gets embodied as different rates of infection vary between the two viruses. (Embodiment is the way life experiences, and the macro- and microsocial forces that shape experience, affect individuals and population health.)

For COVID-19, some of these racially disparate embodiment processes seem (as of this writing in February 2022) to be clear. Given long-standing racist hiring patterns, Black, Latinx, and other racially/ethnically oppressed people are more likely than non-White people to work in industries and occupations that require daily attendance. They undergo increased potential exposure to infection on the way to work and at work. Furthermore, many members of oppressed groups work in industries and occupations whose working conditions facilitate the spread of airborne diseases, such as meatpacking, warehousing, nursing home and home health aide positions, direct service industries such as food service and delivery services, and lower-level hospital positions. Once infected, they return home, where housing and living arrangements may make it impossible to quarantine, so they may expose other members of their family. Further, Black, Latinx, and other oppressed people are disproportionately relegated, through historic housing policies and economic conditions, to crowded housing with poor ventilation, which also contributes to increased SARS-CoV-2 transmission within Black and other oppressed families, and which also reduces their ability to isolate or quarantine after becoming infected in order to protect their families [112]. In addition, racially oppressed groups are greatly over-represented as prisoners, as residents of homeless shelters, and as immigrants detained in what are essentially concentration camps—all of which are places where COVID-19 has spread widely [113,114,115,116,117]. This holds even though some jurisdictions established jail and prison release systems or hotel housing for the homeless that reduced this problem, and some workplaces attempted to improve ventilation or restructured work schedules to reduce transmission risk.

The jobs of members of oppressed racial and ethnic groups often pay very low wages, forcing them to work at two or three jobs, and they often do not provide or offer health insurance or sick day benefits, which makes it much harder for these laborers to gain access to COVID-19 testing or to medical care or to take sick leave. This in turn further increases the likelihood that they will become infected at work, since they are likely to be exposed by infected coworkers facing the same difficulties. Thus, they are more likely to get infected; if infected, they are more likely to get sick, more likely to transmit COVID-19 to their families and to others in their communities, and more likely to die [118,119,120,121,122]. Their COVID-19 fatality rates are also increased by their disproportionate likelihood of having underlying conditions such as obesity, diabetes, or cardiac diseases based both on the high degree of stress associated with being a member of an oppressed group and on racist food industry marketing and retail store zoning and practices that preferentially distribute high fat and sugary foods to oppressed groups [123,124].

The reasons why Black people and other oppressed racial and ethnic groups have higher rates of HIV infection are more indirect and historically embedded, and to some degree less understood, than for COVID-19. The issue is that this pattern (at least for the US, where the data show it clearly) became established early in the HIV epidemic [125,126,127]. Once established, such patterns tend to become self-sustaining (in what is called a founder effect) to the extent that people in an oppressed group tend primarily to have sex with or inject drugs with members of their own group, which is a historic pattern created and maintained by patterns of residential, educational, and occupational segregation and the official and unofficial racism that sustain them. Such segregations tend to restrict opportunities to establish romantic or drug partnerships and are reinforced by the fears oppressed people have of rejection or victimization by the dominant group, as well as by racist cultural tropes such as those against “race mixing.” We should also be clear that there are some racism-produced countertendencies to this pattern of within-group sexual and injection risk-taking. One example is sex work, where Whites use their higher income or other advantages to purchase sexual or other favors from members of the oppressed group. (Our usage of the term “founder effect” here differs somewhat from that used in phylogenetic or molecular evolutionary biology in discussing why certain strains of HIV or another infection become dominant in a given locality).

For the US, at least, there is clear evidence that Black and Latinx people were more likely to be infected than Whites early in the HIV epidemic, and that this was true within each of the major populations with high transmission rates (sexual minority men, people who inject drugs, heterosexuals, and mother-to-child transmission) [128]. This provides evidence that HIV was more prevalent among these oppressed minorities than among non-Whites even in those days when effective HIV treatments were not available. More recent evidence suggests that these patterns of higher HIV prevalence among Black and Latinx populations have persisted into recent years, across multiple transmission risk groups and geographic regions in the US [129].

However, research has not yet identified, or for that matter fully explored, why there was an initial founder effect in which the subordinated and oppressed race/ethnicities were initially more likely to be HIV-infected in the US and some other countries, rather than non-Whites or (in other countries) other dominant groups. It may be that HIV was introduced into these oppressed populations earlier in the epidemic and that risky sex and injection patterns were more likely within racial and ethnic groups than between them; however, there are not sufficient data to disentangle this [125,126,127]. Even if it were discovered that HIV did get introduced initially in oppressed racial and ethnic groups, it would still be important to know *why and how* this came to be the pattern. Clearly, research on this issue is both needed and important, although it will be difficult. Methods of dynamic phylogeographic research such as those used by Vasyleva et al. in studying HIV spread within Ukraine might usefully be applied to this problem [130].

In terms of understanding continued HIV incidence disparities among Black and Latinx populations in the US, one reason for this is simply that HIV incidence tends to be higher in groups with higher HIV prevalence [131,132]. It should also be emphasized that existing research demonstrates that Black and Latinx populations are not more likely to engage in unprotected sex or in syringe sharing during injection drug use, despite the economic inequalities that may make engaging in some safer behaviors (e.g., purchasing condoms and/or new injection equipment) a greater financial burden for these populations. This may in part be because the actual costs of these items are not huge and because many programs make them freely available. It may also be because oppressed groups may adopt both self-protective and altruistic and solidaristic behaviors [133].

Other relationships and processes tend to lead oppressed groups to be more exposed to HIV infection. For example, mass incarceration has disrupted sexual and injection relationships, social networks, and communities disproportionality for non-White groups, which contributes to network turnover and thus to increased HIV risk [134]. Risk networks have been shown empirically to be direct causes of HIV infection [135,136,137]. Further, incarceration is associated with distrust in health care and lack of engagement in primary care, which can affect detection of HIV infection [138].

Members of oppressed groups are also much more likely to not be diagnosed as HIV-positive until the later stages of disease progression [139,140,141]. This means that they are unknowingly HIV infected for a long period of time before being diagnosed and before beginning treatment, both of which can greatly increase the risk of transmission to others within their networks [142].

One reason Black, Latinx, and indigenous people in the US are diagnosed with HIV later in their infections is likely to be the barriers they face in access to health care. They are less likely to have health insurance [143,144,145], less likely to be able to take time off work to visit traditional business-hour health care services, and less likely to be able afford health care and transportation to health care [146,147,148,149,150]. They are also more likely to avoid or otherwise not be engaged in health care, including STI and HIV care, due to a long history of internalized, interpersonal, social, and structural stigmatization targeting marginalized races/ethnicities. Furthermore, this stigma maps onto medical maltreatment [151,152] and could possibly account for differences in the effectiveness of U=U (Undetectable equals Untransmittable) campaigns in reducing anticipated HIV-related stigma [153]. This reduced access to health care also contributes to their having lower access to and use of medications such as pre-exposure prophylaxis (PrEP) to prevent infection or antiretroviral therapy to reduce transmission, both of which contribute to higher infection rates in racially/ethnically oppressed communities.

## 5. How Is Racism Maintained?

Racism has been a strong force in the territory that is now the US for hundreds of years. It has persisted in spite of slave rebellions, a Civil War in which Black people fled in large numbers or took other actions that reduced the ability of the slave-owners to wage war, the organizing and struggles of the Black Reconstruction years (best described by WEB DuBois [154]) and then the Populist uprisings of the 1880s and beyond, and the continuing struggles of Black, Latinx, Asian American, and Native American people throughout the 1900s, with high points in the periods 1956–1980 and then in the years since 2006 [155]. Despite these efforts to eradicate racism, well-documented research shows that structural and institutional racism continue in virtually every institution in US society and that organized racist violence by both official forces like the police and by vigilantes is widespread and murderous [26].

In this final section of the paper, we briefly explore and present some tentative ideas about how and why racism has not disappeared from the US or internationally in spite of centuries of struggle to end it. This might erroneously be considered to be outside the purview of public health as a discipline, though the critical epidemiology of Jaime Breilh [156] and others [157,158,159] would clearly see it as an integral part of epidemiological research and public health action and also see it as requiring new approaches to research frameworks and methodologies. Nonetheless, as our discussion so far indicates, and as the evidence for a vast range of other conditions and diseases has shown, racism continues to be a major cause of health disparities in the US and many other countries [160], including for HIV and other morbidities among people who use drugs [161,162,163,164,165,166]. Thus, we think it is important to consider why this upstream cause of morbidity and mortality (which is not only upstream, since racism manifests itself fairly ubiquitously) persists and to suggest why research is needed about more effective ways to get rid of it.

Our thesis is that racism continues as an organized, multifaceted system for several reasons:
It is extremely useful, perhaps necessary, to the interests and needs of capital and the ruling class. As a corollary to this, it is so deeply intertwined with other core institutions of the system that ending racism will in the best case require massive, rancorous, and likely violent conflict, and that in fact might well be beyond the capacity (and/or interests) of the state and capital to accomplish. We briefly presented some of the evidence for this statement in the section above on history. It is by no means an original analysis [4,52,54,167].Even when state agents and dominant sections of capital try to reduce racism, they do so within the value systems and worldviews of the social order they live in. They tend to see it as a matter of reducing barriers to individual educational attainment and increasing individual opportunities for upward social mobility. Stephen Steinberg [168,169] documented this in great detail for the period starting in the mid-1960s when the moral force and political threat of the Black Movement in the US led political leaders and major capitalist agents, such as the Ford and Rockefeller Foundations, to try to reduce racism [170]. The programs they established did benefit many individual Black people, and they contributed considerably to enlarging the Black middle class as well as ending de jure segregation in the US. This failed, however, to uproot the power and institutional strength of racism because it did not address its institutional roots or the power of those who benefit from racism.A sizable proportion of the economic and political leadership of the world holds racist views of one sort or another. During the period of high colonialism, they were quite overt about this, as the remarks in the History section of this paper about Henry Ford and Herbert Hoover make clear. Since then, many try to hide it or to suppress it within themselves. One previous but still useful analysis on this issue, is Melvin Steinfeld’s *Our Racist Presidents: from Washington to Nixon*, which documented presidents’ various statements and actions of a racist nature. Such racist attitudes are by no means a thing of the past. At the current time, many of the current or recent-past political leaders of the world are overt racists (most notably, perhaps, Trump in the US, Modi in India, and Bolsonaro in Brazil), as are many leaders of capital such as Murdoch and the Koch brothers [171,172,173,174,175,176,177]. Although there have been disagreements over the extent of personal racism among many economic and political leaders, the overt racists have often provided leadership to others who may be more concerned about the importance of racism as a divide-and-rule strategy. The overt racists are also available to encourage and fund a mélange of publications and think tanks that promulgate policies and ideologies that “just happen” to support racialized institutions as well as openly racist ones. Their funding and support is also often offered to grassroots racist organizations, as exemplified by Trump’s support for violently racist demonstrations in Charlottesville and, in India, Modi’s support for violent attacks on non-Hindus.A fourth support for the maintenance (and perhaps expansion) of racial oppression is the widespread individual racism that exists in many countries (and is often normative within some families and communities and institutionalized in linguistic patterns). To some extent, this is the result of the deliberate scapegoating and stigmatization that is an important part of elites’ divide-and-rule strategies. More important, perhaps, is the ideological impact of living in a heavily racialized, competitive, and putatively meritocratic society. As discussed above, living and working in such an environment encourages Whites to see themselves as superior and other socially constructed races and ethnicities as subordinate and inferior. This also justifies, in their minds, their White children getting the best schooling, jobs, and housing. It also justifies, in their minds, the violent policing of racially and ethnically oppressed people and of many members of oppressed groups being relegated to poor housing and education focused, if anything, on reproducing a low-wage, disposable workforce. Efforts to reduce or eliminate racism get interpreted, in this context, as attacking the “well-earned rights” of Whites and what Du Bois referred to as the “psychological wage of Whiteness” [154]. The result is both a powerful White electoral constituency against legislative or other efforts to reduce racist structures and norms and a strong base for vigilante or official violence against racialized populations and/or their supporters.


This individual racism has many structural supports. Racist educational systems provide minimal and biased accounts of history and of current racial and ethnic realities. Mass media and other institutions, as discussed above, further promulgate racist images and cultural tropes. In addition, the dominance of White (and other oppressor) people, their history, and their much lauded “western culture” is strengthened by, and has contributed to, racism embedded in language and language policies and by stigmatizing the speech patterns of Blacks, Latinx, and other oppressed groups [178,179,180].

## 6. Conclusions

This paper has presented evidence that the social structures and processes of racism, in particular but not limited to anti-Black racism, led and continue to lead subordinated, oppressed racial/ethnic groups, including those among the key populations of PWUD, men who have sex with men, and heterosexual populations, to be more adversely impacted by HIV (and COVID-19) epidemics than Whites. This paper has primarily focused on anti-Black racism as a lens to encapsulate the experience of all oppressed racial, ethnic, and other oppressed groups. It discusses how modern racism developed out of the interests and the divide-and-rule strategies of rich and powerful White people and describes the ways in which racism is organizationally structured from the state, national, and international levels to the daily activities of workers who (consciously, unconsciously, and/or reluctantly) apply the rules and norms that result in Black people, Latinx people, and other oppressed groups being harmed in numerous ways. It discusses the mechanisms through which disease disparities get produced. Finally, it discusses why racism is a resilient structure that has so far remained strong despite centuries of being attacked.

This paper has not focused on the differences among the various forms that racial and ethnic oppression take in the United States or elsewhere. Such differences stem from different patterns of previous conquest and enslavement and on preexisting cultures and their adaptations and evolutions. Thus, the oppression of Blacks in the US, of different ancestry Latinx subgroups in the US, and of Muslim peoples in neoliberal India (as just one of many non-US examples) vary in many ways. These varied oppressions in particular contexts can adversely affect PWUD including raising HIV or hepatitis infection rates and subsequent morbidity and mortality and lead to differences such as those among Latinx ethnic subgroups in drug overdose fatality rates in the US [181,182,183,184,185].

What does this mean for public health and for public health researchers? First, it poses an unavoidable task. The overall contributions of structural racism to disease, morbidity, and mortality are both vast and deeply unfair, as well as being potentially (but not easily) changeable. This poses an ethical duty and practical need to conduct research and engage in action to undo racist structures and beliefs. It further requires attention on the part of researchers to ensure that research questions and methods themselves are not structurally racist, or do not inadvertently assume or reinforce racist attitudes and structures [159].

Given the wealth and power of the supports for racism, this will not be easy, but it has been done. Even as late as 1955, most social scientists agreed that de jure racist structures such as Jim Crow in the South of the US or apartheid in South Africa were not vulnerable to successful attack. Nonetheless, mass movements of the oppressed and their allies succeeded in ending both Jim Crow and apartheid, although racism as a larger de facto system remains intact.

With respect to research and teaching, this means that we need to study the vulnerabilities both of the racist system and of the systems and groups that support it, just as researchers are studying the vulnerabilities of cancer cells and of the systems that provide them with nutrients or make them virulent and malignant. In part, we can do this through studying the history of prior challenges to racist systems to identify why some were partial successes and why others were not. What were their methods? Who took part? How were they recruited, mobilized, and organized? How effective were they in what domains and for what durations? How well did they address issues of intersectional oppressions of race, ethnicity, class, gender, and sexual identity? Since capital and the state, and their divide-and-rule strategies, are important supports for the maintenance of racism, what are their vulnerabilities? How can they be addressed?

With respect to antiracist strategy, this paper demonstrates that addressing racism is not as simple as having people take some courses on micro-aggression and implicit racism. Nonetheless, since individual racism makes it harder to eliminate institutionalized racism or the institutional power of the capitalism and the rich that supports it, research on new ways of communication or other action that can address this at the population level could be useful. We stress the importance of seeing this in terms of population-level impacts rather than putting hope in programs that work on one individual at a time, since these are both too slow to have an appreciable affect and are counteracted by the normative influence of a racially structured reality. One hypothesis is that social movements such as the movements for Black lives of recent years or social revolutions are the most effective ways to weaken individual racism as well as being ways to attack racism as a system of power and oppression [186,187,188,189,190,191,192].

Finally, from our studies and research, it will take (at a minimum) large-scale social movements with firm roots in the communities and workplaces of the oppressed, and it will probably require respectful and collaborative cooperation with other oppressed groups and with other movements and other oppressed groups (for workers health, safety, and power, and for environmental sustainability, and for social justice and new life opportunities for women and sexual minorities, at a minimum).

None of this will be easy. Much of it will be risky. It will involve political action, as does much of public health. But it is necessary.

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
