# Peer review of "Toward a Theory of the Underpinnings and Vulnerabilities of Structural Racism: Looking Upstream from Disease Inequities among People Who Use Drugs"

_ijerph, 2022, doi:10.3390/ijerph19127453_

Round 1
Reviewer 1 Report
Friedman and colleagues have drafted an excellent review on racial oppression and HIV among people who inject drugs and summarised related theories to support the content. The manuscript is well written, please find few minor suggestions.
- Line 51, line 59: the references are unclear, please revise references
- The authors have provided international context of the subject for e.g. line 84-96. However, in the subsequent sections the internationalisation of the topic is superficial. I would recommend the authors to add content to this.
- A separate recommendation section based on the evidence would be easier to grasp for the readers and policy makers.
- This review looks very comprehensive, I acknowledge significant numbers of references are required. However, if possible, I would recommend the authors to limit the references as the count is 186.
Author Response
Reviewer 1
Friedman and colleagues have drafted an excellent review on racial oppression and HIV among people who inject drugs and summarized related theories to support the content. The manuscript is well written, please find few minor suggestions.
- Thank you for your support.
- Line 51, line 59: the references are unclear, please revise references
The two places that were confusing were “calls” for the footnotes at the bottom of the page.
- The authors have provided international context of the subject for e.g. line 84-96. However, in the subsequent sections the internationalisation of the topic is superficial. I would recommend the authors to add content to this.
We think our re-organization and additions do this reasonably well. However, page limits, and the available data, limit how much detail we can give on this and so we have made more explicit in the Introduction (lines 45-46) that the focus of the paper is on the US.
- A separate recommendation section based on the evidence would be easier to grasp for the readers and policy makers.
We have re-named the Conclusions as “Conclusions and Recommendations.” Most of this section is recommendations as it stands. Unfortunately, there are no simple solutions to structural racism, and we have framed the action issues as we understand them in this section.
- This review looks very comprehensive, I acknowledge significant numbers of references are required. However, if possible, I would recommend the authors to limit the references as the count is 186.
Here, we will defer to the editors’ judgement. We prefer not to do this, since we think the references add a lot to the value of an article such as this.
Reviewer 2 Report
The research title is too long, make it concise.
The abstract does not mention the type of analysis that has been performed.
Remove 'people who uses drugs' from the keyword list.
There is not enough evidence to show the racism impact has on HIV/AIDS. Some historical moments can be shared here.
The footnote is too long. The author should try to incorporate it within the main text.
There is no clarity why this research is needed. The authors tried to put so much information in the background. Try to make it more specific to the research problem.
Is this a conceptual paper or a review? if it's a review what kind of review method is followed?
Author Response
Reviewer 2
The research title is too long, make it concise.
- We have retitled it in ways that help clarify what the paper is and is not.
The abstract does not mention the type of analysis that has been performed.
- We have clarified that the paper is a conceptual review.
Remove 'people who uses drugs' from the keyword list.
- Although the focus is upstream, the paper does contain a review of racial inequities and their effects on the health of people who use drugs. We thus prefer to retain this keyword.
There is not enough evidence to show the racism impact has on HIV/AIDS. Some historical moments can be shared here.
- We agree and have added text directly linking sections to show the impact of racism on HIV/AIDS (see, for example, lines 194-197, 356-362)
The footnote is too long. The author should try to incorporate it within the main text.
- We have shortened it. Our past experience finds that this kind of material works best as a footnote.
There is no clarity why this research is needed. The authors tried to put so much information in the background. Try to make it more specific to the research problem.
- We have tried to clarify why it is needed. Most importantly, we have added that “downstream” interventions have not solved the issue of racial/ethnic health inequities, and thus that public health should try to understand structural racism, its institutionalization, and why it has not been ended in spite of a great deal of human effort over many years. We have also clarified that this paper is a conceptual review rather than an empirical analysis of the standard epidemiologic
Is this a conceptual paper or a review? if it's a review what kind of review method is followed?
- It is a conceptual review
Reviewer 3 Report
The authors described ins review some ideological issue about the concept of racism. They organized the manuscript to emphasize the rules of racism. In my opinion, the review was ell organized. It is suitable to publish in International Journal of Environmental Research and Public Health. However, major post must be addressed and posed on. English grammar structure and English language need a revision. The authors must check the manuscript layout as required by the publication policy.

Author Response
Reviewer 3
We appreciate the praise of the article, and have tried to adjust flaws in grammar and style. We are not sure what you meant by “ However, major post must be addressed and posed on,” but suspect that we have responded in the other changes we have made.
Reviewer 4 Report
Dear Author,
thank you for your interesting article, which I found inspiring for my own field of interest. I have some issues I suggest addressing. First, some suggestions regarding written English:
- Pay attention to abbreviations, which must be explained before using them. For example, you use both PWID and PWUD and you explain only PWUD.
- Page 2, line 46: “mortality related to AIDS” should be “AIDS-related mortality”.
- Page 2, lines 51-52: “Racial and ethnic disparities in health and disease long pre-dated HIV and include differential rates of SARS CoV-2 (etc)”. Since you start writing that disparities long pre-dated HIV, I would not start with SARS-CoV-2. I would rather put it at the end with a sentence like “ and the ongoing pandemic of SARS-CoV-2” or something like that.
- Page 2, line 59-61: when you mention all the prevalence for ethnicity, I would also mention the prevalence for White people to be able to make a comparison.
- Page 2, lines 67-70: I think this sentence is not clear and not completely correct. I would explain further what you mean by intersection with gender and class. I guess you mean intersection with gender and class-based discrimination, but I am not sure.
- Page 3, line 81: “including HIV and…”, this sentence is incorrect. I would rather write “by raising the risk of contracting HIV and…” or something like that.
- Regarding the chapter “Description of disparities”, I would cite the literature from other countries first, and then disclose that you will focus only on an ethnic group in one country and, in particular, on a high-risk subgroup (PWID), and you will not focus on different forms of racism and oppression. Is to say, regarding the last paragraph I would displace “This paper does not focus on the differences among the various forms that racial and ethnic oppression takes. Such differences stem from different patterns of previous conquest, enslavement, and on pre-existing cultures and their adaptations and evolutions. Thus, the oppression of Blacks in the US, of different ancestry Latinx subgroups in the US, and of Muslim peoples in neoliberal India (as just one of many non-US examples) vary in many ways” to the end.
- Regarding the sentence “These varied oppressions in particular contexts can adversely affect people who use drugs (PWUD) including HIV or hepatitis infection rates and subsequent morbidity and mortality. They also affect overdose rates. For example, drug overdose fatality rates vary greatly among different Latinx subgroups in the US (24-28).”, I do not think it fits here. I think we are discussing the fact that racism is a determinant for health inequities in many countries. This sentence is more about the first part of the paragraph, in which you discuss how racism impacts on health access in the US.
- Page 3, lines 100-105: I suggest presenting this into-graph sentence as an end-page note.
- Page 5, last line: there is a repetition of the word ”mine”.
- Page 10, lines 4004-441: “One reason for Black, Latinx, and indigenous people in the US get diagnosed with HIV later in their infections is likely to be the barriers they face in access to health care.” This sentence is not clear, I suggest rephrasing.
- I think that the chapter regarding the maintenance of racism should be placed before the chapter regarding embodiment.
Regarding contents:
- I think the title does not reflect at all the intent of the article, since a very brief part is dedicated to disparities in HIV prevalence among PWID. Moreover, the focus of the whole article is racism against Black and Latinx in US and how it determines health inequities. I do not understand why focusing on ethnicity-determined health inequities among PWID, when this area is barely mentioned in the article.
- I think the overview regarding history of racism and its institutionalization is very interesting, yet disproportionately long in respect with the part dedicated to health inequities. As it is written, I think this article is more suitable for other kind of journals. It would be interesting to publish if the part regarding health inequities would be more accurate.
- I would go deeper in the paragraph “Embodiment”. You explain very well reasons for inequities during COVID-19 pandemic. However, I think different reasons contribute to delayed access to care as far as HIV and sexually transmitted infections are concerned. There is an important role for stigma and self-stigma which could lead to fear of discrimination and further marginalization that should be discussed.
Author Response
Reviewer 4
Thank you for your points 1 through 12 on our wording and organization. We have accepted almost all of these and made the appropriate corrections.
We also greatly appreciated your suggestions on substance (content), and have responded as follows:
On #1, we have changed the title to read “Towards a theory of underpinnings and vulnerabilities and supports of structural racism: Looking upstream from disease inequities among people who use drugs”.
On #2, where Reviewer 4 says: I think the overview regarding history of racism and its institutionalization is very interesting, yet disproportionately long in respect with the part dedicated to health inequities. As it is written, I think this article is more suitable for other kind of journals. It would be interesting to publish if the part regarding health inequities would be more accurate.
We think the section on history and institutionalization is critical to the points being made. As we have clarified in response to other reviewers’ suggestions, the focus of this paper is on the “upstream,” since the effects of racism on health have proven to be highly resistant to downstream efforts to resist them and thus the field needs to study and understand the upstream factors better. We are not sure what inaccuracy is being referred to, but we have corrected those mentioned in the reviews.
On #3, which suggests that our discussion of embodiment was insufficient, we want to thank Reviewer 4 deeply. We totally agree with this point, and are very embarrassed to miss it, because most of our team recently wrote a paper on “The Stigma System” that sparked our interest in writing this paper. We have added the following: “They are also more likely to avoid or otherwise not be engaged in health care, including STI and HIV care, due to a long history of interpersonal, internalized and social stigmatization of members of oppressed race/ethnicities and due to medical maltreatment towards them in a number of well-documented instances. “ (With appropriate references)
Round 2
Reviewer 3 Report
The authors improved the manuscript adding new some elements in all the sections.
In my opinion the paper is actually suitable to IJERPH publication. However, the authors must editing English grammar structure.
Q1. Lane 58 page 2……. who injected or injects (e.g). Check all typing mistakes.
Q2. The authors must check the bibliographic styles.

Reviewer 4 Report
Dear authors,
Thank you for your reply to my previous revision. Reading again the manuscript, I was wondering if racism and inequities could have an impact about the U=U message in this population. I suggest you to add a sentence about U=U in your paper, because in my opinion this message should be reported in a paper like yours; I think that the promulgation of U=U message could help to reduce the stigma in people living with HIV. If you do not have enough information about it I suggest you some paper that you could read and cite (10.1097/QAD.0000000000002825; 10.1016/S0140-6736(19)30418-0)
